# Dengue severity by serotype and immune status in 19 years of pediatric clinical studies in Nicaragua

Federico Narvaez[1,2], Carlos Montenegro[2], Jose G. Juarez[2], José Victor Zambrana[2,3], Karla Gonzalez[2,4], Elsa Videa[2], Sonia Arguello[2], Fanny Barrios[1,2], Sergio Ojeda[2], Miguel Plazaola[2], Nery Sanchez[2], Daniel Camprubí-Ferrer[5¤], Guillermina Kuan[6], Gabriela Paz Bailey[5], Eva Harris[7], Angel Balmaseda [2,4]*

1 Unidad de Infectología, Hospital Infantil Manuel de Jesús Rivera, Ministerio de Salud, Managua, Nicaragua, 2 Sustainable Sciences Institute, Managua, Nicaragua, 3 Department of Epidemiology, School of Public Health, University of Michigan, Ann Arbor, Michigan, United States of America, 4 Laboratorio Nacional de Virología, Centro Nacional de Diagnóstico y Referencia, Ministerio de Salud, Managua, Nicaragua, 5 Dengue Branch, Division of Vector-Borne Diseases, National Center for Emerging and Zoonotic Infectious Diseases, San Juan, Puerto Rico, 6 Centro de Salud Sócrates Flores Vivas, Ministerio de Salud, Managua, Nicaragua, 7 Division of Infectious Diseases and Vaccinology, School of Public Health, University of California, Berkeley, California, United States of America

¤ Current address: ISGlobal, Hospital Clínic - Universitat de Barcelona, Barcelona, Spain
* abalmaseda40@gmail.com

## Abstract

### Background

Dengue virus, a major global health threat, consists of four serotypes (DENV1-4) that cause a range of clinical manifestations from mild to severe and potentially fatal disease.

### Methods

This study, based on 19 years of data from the Pediatric Dengue Cohort Study and Pediatric Dengue Hospital-based Study in Managua, Nicaragua, investigates the relationship of serotype and immune status with dengue severity. Dengue cases were confirmed by molecular, serological, and/or virological methods, and study participants 6 months to 17 years old were followed during their hospital stay or as ambulatory patients.

### Results

We enrolled a total of 15,833 participants, of whom 3,308 (21%) were positive for DENV infection. Of 2,644 cases with serotype result by RT-PCR, 559 corresponded to DENV1, 1,002 to DENV2, 760 to DENV3 and 323 to DENV4. Severe disease was more prevalent among secondary DENV2 and DENV4 cases, while similar disease severity was observed in both primary and secondary DENV1 and DENV3 cases. According to the 1997 World Health Organization (WHO) severity classification, both DENV2 and DENV3 caused a higher proportion of severe disease compared to other serotypes, whereas DENV3 caused the greatest percentage of severity according to the WHO-2009

**Data availability statement:** After securing approval from the UC Berkeley Committee for the Protection of Human Subjects, individual data for figure reproduction can be shared with external researchers. For data access arrangements, please contact parasitologia@minsa. gob.ni (Point of Contact: Alberto Montoya Perez, IRB coordinator at CONIS, Nicaragua) or the CPHS at ophs@berkeley.edu. Standard data transfer agreements govern all data used in this study. The associated code is available at the following link: https://github.com/jgjuarez/ Narvaez_Dengue_Clinical_Severity.

**Funding:** This work was supported by the National Institute for Allergy and Infectious Diseases (NIAID) of the National Institutes of Health (NIH) grant numbers P01 AI106695 (EH), R01 AI099631 (AB), U01 AI153416 (EH), U19 AI118610 (EH), U54 AI65359 (Barbour; subcontract AB), and BAA-NIAID-DAIT-NIHAI2009061 (Loeb; subcontract AB) from NIAID/NIH, and the Pediatric Dengue Vaccine Initiative grant VE-1 (EH) and the FIRST grant (EH; Jaime Sepulveda, PD) from the Bill and Melinda Gates Foundation. The funders had no role in study design, data collection and analysis, decision to publish, or preparation of the manuscript.

**Competing interests:** The authors have declared that no competing interests exist.

classification. DENV2 was associated with increased odds of pleural effusion and low platelet count, while DENV3 was associated with both hypotensive and compensated shock.

## Conclusions

These findings demonstrate differences in dengue severity by serotype and immune status and emphasize the critical need for a dengue vaccine with balanced effectiveness against all four serotypes, particularly as existing vaccines show variable efficacy by serotype and serostatus.

## Author summary

Dengue, a major public health threat, is a viral infection spread by mosquitoes that can cause a range of symptoms from mild to life-threatening. Our study analyzed 19 years of data from two pediatric studies of dengue in children 6 months to 17 years of age in Managua, Nicaragua. We aimed to understand how dengue severity varies with different serotypes of dengue virus (DENV1-4) and whether the child had experienced a previous DENV infection or not. We also used two classification schemes of dengue severity, published by the World Health Organization in 1997 and 2009. Overall, we found that DENV2 and DENV3 caused the most severe disease in children. DENV3 and DENV1 led to severe outcomes in both primary (first) infection and secondary infections with another DENV serotype, while DENV2 and DENV4 manifested severity in secondary infections. These findings highlight the importance of a balanced dengue vaccine that can protect against all four virus serotypes, as current vaccines show varying effectiveness by serotype. Our research emphasizes the need for effective prevention and treatment strategies to manage dengue, particularly in regions where the virus is widespread.

## Introduction

The four serotypes of dengue virus (DENV1-4), a mosquito-borne *Flavivirus*, affect tens of millions of people worldwide [1]. In recent decades, dengue epidemics have consistently increased in tropical and subtropical regions of the world, with vector control as the principal means of prevention due to the lack of an effective, widely used vaccine. Dengue viruses cause a range of clinical manifestations, from mild to severe and potentially fatal [2]. Understanding the contribution of serotype and immune status (primary versus secondary DENV infection) to clinical spectrum and severe disease is important for developing effective prevention and control strategies, particularly when introducing new vaccines with differential efficacy [3,4].

Developing a dengue vaccine has been challenging, primarily because it needs to provide protection against all four DENV serotypes, as imbalanced protection may result in antibody-dependent enhancement of infection and disease [5,6]. The two approved dengue vaccines vary in efficacy by serotype and immune status. Specifically, Dengvaxia shows low efficacy for DENV2 and results in increased risk of hospitalization when administered to DENV-naïve recipients [7]; QDenga has low efficacy against DENV3 in dengue-naïve children and unknown efficacy against DENV4 [8]. Thus, understanding how dengue severity is modulated by serotype and immune status is critical for public health policy makers. Several studies have addressed this question, most using the 1997 World Health Organization (WHO)

severity classification, with only a handful using the 2009 WHO definitions for severe dengue (SD) [9–13]. These studies showed that both DENV2 and DENV3 are associated with the greatest clinical severity [10,14–17]. A previous analysis of our hospital-based study in Nicaragua also showed that DENV2 cases had a higher frequency of shock and internal hemorrhage when compared to DENV1 cases [18]. However, a comprehensive understanding of how dengue severity varies by serotype and immune status remains limited, especially in Latin America compared to other regions like Southeast Asia [10,15–17].

For the past 19 years, we have conducted studies on the diagnosis, classification, and clinical management of dengue in Managua, Nicaragua [18–23]. Here we describe the clinical features and severity of dengue cases stratified by serotype and immune status in almost 3,000 patients presenting to our study health center or hospital and followed over the entire course of illness. We analyze primary and secondary cases of all four DENV serotypes classified by both the 1997 and 2009 WHO guidelines.

## Methods

### Ethics statement

The Pediatric Dengue Cohort Study (PDCS; 2004 to present) and the Pediatric Dengue Hospital-based study (PDHS; 2005 to present). were approved by the Institutional Review Boards of the Nicaraguan Ministry of Health (Protocol #CIRE-09/03/07-008.Ver24 and CIRE-01/10/06-13.ver8, respectively) and the University of California Berkeley Committee for the Protection of Human Subjects (Protocols 2010-09-2245 and 2010-06-1649, respectively). Written informed consent was obtained from all parents or legal guardians before enrollment; assent was obtained from children 6 years of age and older.

### Study design

We leveraged two ongoing studies in Managua: PDCS and PDHS. The PDCS is the longest ongoing prospective dengue cohort study, currently in its 20th year, following ~4,000 children aged 2–17 years old (y/o) in a community-based context, with enhanced passive surveillance at the Health Center Sócrates Flores Vivas (HCSFV) in District 2 of Managua [24]. To maintain the age structure of the PDCS, 200–300 new 2–y/o children are enrolled every year, along with an additional ~200 children 3–7 y/o to compensate for loss to follow-up [24]. The PDHS is a clinical study that enrolls patients every year during the dengue season at the Hospital Infantil Manuel de Jesus Rivera (HIMJR), Nicaragua's national pediatric reference hospital. The PDHS enrolls children from 6 months to 14 y/o with symptoms and signs suggestive of an arbovirus infection (WHO 1997) [25] either in the emergency room or in the Infectious Diseases Unit of the HIMJR [20]. Children in the PDCS who require hospitalization are referred to the HIMJR, including patients 15–17 y/o. All participants were followed for their entire course of illness. Both the HCSFV and the HIMJR provide free health care services. Here, we present information from patients enrolled in both studies from September 1, 2004, to February 18, 2024.

### Classification of dengue disease severity

Dengue disease severity was defined by the World Health Organization (WHO) guidelines of 1997 (Dengue Fever [DF], Dengue Hemorrhagic Fever [DHF], Dengue Shock Syndrome [DSS]) [25] and 2009 (Dengue without Warning Signs [DwoWS], Dengue with Warning Signs [DwWS] and Severe Dengue [SD]) [2]. See Table A in S1 Text for more details. Here, we defined severe dengue disease as DHF/DSS (1997) or SD (2009).

### Laboratory methods and dengue diagnostics

Cell blood counts (CBC) and blood chemistry tests were conducted using a Cell/Dyn Rubi (Abbot) and Biosystems BA 400 (Biosystems) automated system, respectively, and the trend over time in each patient's platelet and hematocrit values was reviewed. Dengue cases were confirmed by: 1) RT-PCR/viral isolation in acute-phase (days 1–6 post-onset of symptoms) samples and/or 2) seroconversion by DENV IgM MAC-ELISA and/or seroconversion or a ≥4-fold increase in total DENV antibody titers as measured by the Inhibition ELISA (iELISA) in paired sera from the acute phase and early convalescent phase (14–28 days post-onset of symptoms) [26,27]. Viral RNA was initially detected using a nested or semi-nested RT-PCR [28,29] and later using a multiplex Zika-Chikungunya-Dengue (ZCD) real-time RT-PCR (rRT-PCR) [30] followed by serotyping of DENV-positive samples by multiplex DENV rRT-PCR [31]. Immune status was determined using iELISA in convalescent samples; <2,560 was considered primary infection and ≥2,560 was considered secondary infection [19,26]. Patients for whom it was not possible to identify the immune response were excluded from the analysis.

### Statistical methods

Descriptive statistics were used to summarize demographic and dengue case data from the PDCS and PDHS, yielding participant counts, serotype distribution, and incidence data (see Figs A–J in S1 Text for a breakdown of all results by study). Incidence and distribution of infections by serotype and year were calculated. In addition, we analyzed severity by serotype using both the 1997 and 2009 WHO classifications, reporting the distribution of primary and secondary DENV infections and the severity classifications by each serotype in raw numbers and percentages. We evaluated the odds of secondary infections by serotype given severe DENV infection, and p-values were calculated using chi-square test or fisher test. To analyze severity based on the 1997 and 2009 WHO dengue classifications by serotype and immune status, we used marginal effects from multivariate logistic regressions, producing predictive percentages of severe cases by serotype and immune status and applying an interaction term by serotype and immune status, controlling for age and sex.

   We implemented separate logistic regression models adjusting by immune status to calculate the odds ratios of each symptom and sign by serotype, using DENV1 as the reference value. The signs and symptoms comprised a range of clinical indicators including hypotensive shock, compensated shock, poor capillary refill, pleural effusion, hemoconcentration, low platelet count, mucosal bleeding, intensive care unit (ICU) admission, inotropic drug usage, abdominal pain, vomiting, rash, headache, and myalgia. All analyses were performed in R version 4.2.2 (R core Team). All the analyses and figures were also stratified by study (PDCS and PDHS) (Figs B–J in S1 Text).

## Results

### Demographic data and dengue cases

This study includes 15,833 participants, of whom 3,308 (21%) were positive for DENV infection. Yearly enrollment (PDHS) or community-based suspected dengue cases (PDCS) varied depending on the epidemic season, with a maximum of 1,806 children in 2019, a minimum of 207 children in 2004, and an average of 754 (Standard Deviation = 392) children per year combining both studies. Of the 3,308 dengue cases, 2,752 (83%) were DENV-positive by RT-PCR. Of these, the immune response was determined in 2,644 (96%) patients; only the latter were analyzed. As a result, 559 DENV1, 1,002 DENV2, 760 DENV3, and 323 DENV4 cases were identified (Table 1). Children were distributed evenly by sex, with 1,318 (49.8%) females, and

**Table 1. Study population: demographic information, DENV serotype, and health facility, PDCS and PDHS, Managua, Nicaragua, 2004–2024.**

|  | PDCS (n = 1300) | PDHS (n = 1,344) |
| --- | --- | --- |
| **Age** | 9.61 (SD = 3.60) | 9.01 (SD = 3.76) |
| 6–11month | 0 (0%) | 31 (2.3%) |
| 1 to 5 y | 171 (13%) | 236 (18%) |
| 6 to 10 y | 535 (41%) | 523 (39%) |
| 11 to 14 y | 465 (36%) | 551 (41%) |
| 15 to 17 y | 129 (10%) | 3 (0.2%) |
| **Sex** | | |
| F | 670 (52%) | 648 (48%) |
| M | 630 (48%) | 696 (52%) |
| **Serotype** | | |
| DENV1 | 291 (22%) | 268 (20%) |
| DENV2 | 470 (36%) | 532 (40%) |
| DENV3 | 309 (24%) | 451 (34%) |
| DENV4 | 230 (18%) | 93 (7%) |

the average age of disease onset was 9.88 y/o in the PDCS and 9.19 y/o in the PDHS (Table 1). Children were followed throughout the course of disease and were managed as inpatients (mean hospitalization = 3.9 days) or outpatients (mean follow-up = 2.5 days) depending on disease severity. Of the 1,300 DENV-positive participants in the PDCS, 441 (34%) required hospitalization following national guidelines and were transferred to the study hospital.

## DENV serotype and immune status

Over the 19 years of the studies, we observed varying patterns of circulating DENV serotypes. DENV2 and DENV3 were the most common serotypes, with predominantly secondary infections for DENV2 and similar numbers of primary and secondary infections for DENV3 (Fig 1 and Fig A in S1 Text). Similar trends were observed in the PDCS and PDHS (Fig B in S1 Text). Interestingly, after 5 years of only DENV2 cases followed by 2 years of COVID-19 travel restrictions, in 2022, we experienced epidemic levels of DENV4 for the first time in 30 years and simultaneous circulation of all four serotypes. Overall, we observed 919 primary infections and 1,725 secondary infections, with primary infections being significantly more frequent in the ages of 6–10 years and secondary infections significantly more prevalent in ages 11–14 years, with an almost equal distribution by sex (Table 2).

## Dengue severity by immune status and serotype

The clinical spectrum caused by each serotype was analyzed based on the 1997 and 2009 WHO definitions [2,25]. Using the 1997 classification, 2,228 (84%) patients were DF and 416 (16%) were DHF/DSS (Table 3). Here, severity was defined as DHF/DSS. Using WHO-2009, 1,530 (58%) patients were classified as DwWS and 436 (17%) with SD; therefore, severity was defined as SD (Table 3). Evaluating immune status, we observed that secondary cases were much more prevalent in severe disease caused by DENV2 infections (WHO-1997: 207 [94%] and WHO-2009: 125 [89%]) and DENV4 infections (WHO-1997: 9 [100%] and WHO-2009: 9 [82%]), while for DENV1 (WHO-1997: 27 [64%] secondary and WHO-2009: 50 [59%] secondary) and DENV3 (WHO-1997: 88 [61%] secondary and WHO-2009: 101 [50%] secondary), disease severity was observed in both primary and secondary infections (Fig 2).

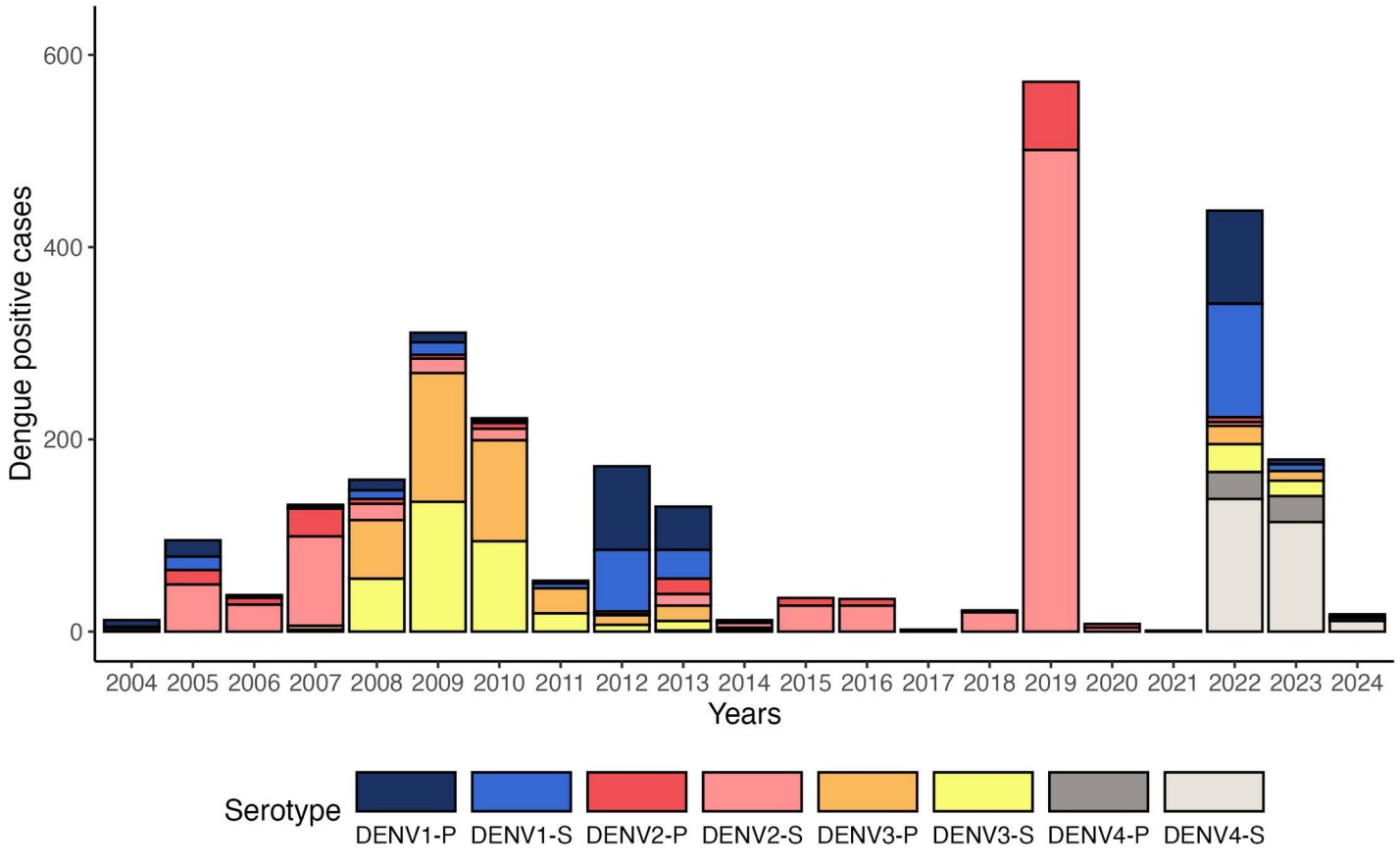

**Fig 1. DENV circulation by immune status in the PDCS and PDHS combined, Managua, 2004–2024.** Circulating serotypes and immune status in dengue cases in the studies in Managua are represented. The hospital study was not run in 2020 and 2021. P, primary; S, secondary.

**Table 2. Age distribution of dengue cases by DENV serotype and immune status, PDCS and PDHS, Managua, Nicaragua, 2004–2024.**

| Serotypes | DENV1 | | DENV2 | | DENV3 | | DENV4 | |
|---|---|---|---|---|---|---|---|---|
| | 1° | 2° | 1° | 2° | 1° | 2° | 1° | 2° |
| Age group* | | p<0.001 | | p<0.001 | | p = 0.001 | | p = 0.007 |
| 6–11 months | 6 (2.1%) | 0 (0%) | 8 (4.7%) | 3 (0.4%) | 12 (3.1%) | 2 (0.5%) | 0 (0%) | 0 (0%) |
| 1 to 5 years | 57 (20%) | 11 (4.1%) | 46 (26%) | 93 (11%) | 146 (38%) | 40 (11%) | 6 (10%) | 8 (3.0%) |
| 6 to 10 years | 122 (42%) | 103 (38%) | 86 (47%) | 299 (36%) | 172 (44%) | 171 (46%) | 24 (41%) | 81 (31%) |
| 11 to 14 years | 100 (35%) | 140 (52%) | 38 (21%) | 389 (47%) | 55 (14%) | 145 (39%) | 17 (29%) | 132 (50%) |
| 15 to 17 years | 4 (1.4%) | 16 (5.9%) | 4 (2.2%) | 36 (4.4%) | 4 (1.0%) | 13 (3.5%) | 12 (20%) | 43 (16%) |
| Sex* | | p > 0.9 | | p = 0.5 | | p = 0.7 | | p = 0.7 |
| F | 136 (47%) | 128 (47%) | 93 (51%) | 395 (48%) | 194 (50%) | 191 (51%) | 35 (60%) | 146 (55%) |
| M | 153 (53%) | 142 (53%) | 89 (49%) | 425 (52%) | 195 (50%) | 180 (49%) | 24 (40%) | 118 (45%) |

*Fisher's Exact test was used for statistical comparisons to evaluate association between primary and secondary infection within each serotype by age and sex.

**Table 3. Dengue severity by serotype and immune status using the WHO 1997 and 2009 classifications, PDCS and PDHS, Managua, Nicaragua, 2004–2024.**

| Serotypes | DENV1 | | DENV2 | | DENV3 | | DENV4 | |
|---|---|---|---|---|---|---|---|---|
| | 1° | 2° | 1° | 2° | 1° | 2° | 1° | 2° |
| **WHO-1997** | | | | | | | | |
| **DF** | 262 (95%) | 239 (90%) | 164 (92%) | 610 (75%) | 331 (86%) | 282 (76%) | 57 (100%) | 249 (97%) |
| **DHF** | 12 (4.3%) | 21 (7.9%) | 12 (6.7%) | 159 (20%) | 43 (11%) | 72 (19%) | 0 (0%) | 9 (3%) |
| **DSS** | 3 (1.1%) | 6 (2.3%) | 2 (1.3%) | 48 (5.9%) | 13 (3.4%) | 16 (4%) | 0 (0%) | 0 (0%) |
| **WHO-2009** | | | | | | | | |
| **DWoWS** | 97 (35%) | 90 (34%) | 74 (42%) | 106 (13%) | 72 (18%) | 75 (20%) | 29 (51%) | 101 (40%) |
| **DWWS** | 146 (53%) | 126 (48%) | 89 (50%) | 586 (72%) | 216 (56%) | 194 (53%) | 25 (44%) | 148 (57%) |
| **SD** | 34 (12%) | 50 (19%) | 15 (8%) | 125 (15%) | 99 (26%) | 101 (27%) | 3 (5%) | 9 (3%) |

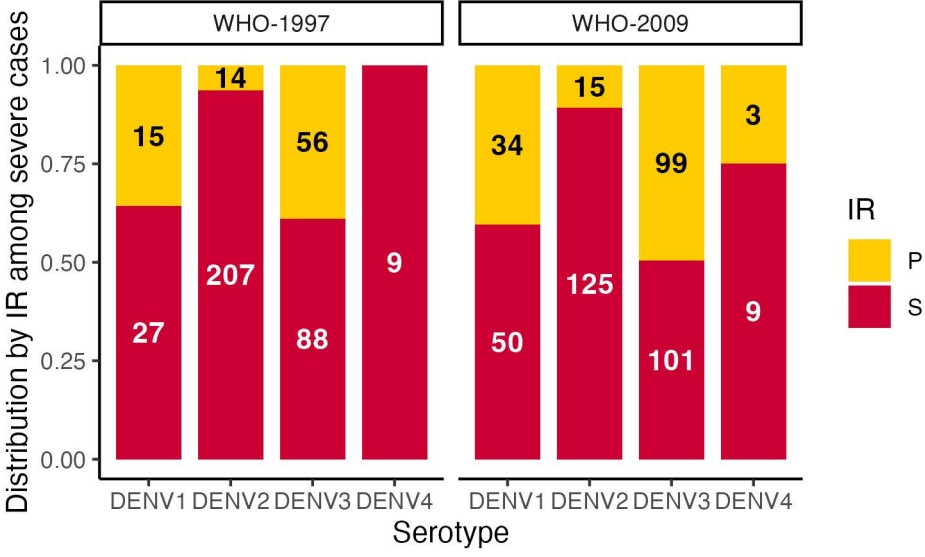

**Fig 2. Distribution by immune response among severe dengue cases, PDCS and PDHS combined, Managua, 2004–2024.** P, primary; S, secondary; IR, immune response.

The breakdown of severe disease by immune status of DENV1 and DENV3 was significantly different from that of DENV2 (DENV2-DENV1: OR 8.21, 3.58–18.87; DENV2-DENV3: OR 9.41, 4.98–17.78). Similar trends were observed overall in the PDCS and PDHS (Fig C in S1 Text). Evaluating by serotype, we found that DENV3 was associated with the greatest severity in both primary and secondary infections according to both 1997 and 2009 WHO classifications, while DENV2 was most associated with DHF/DSS in secondary cases (model-derived estimates compared to DENV1 in Figs 3 and D.A in S1 Text). Additionally, when adjusting for epidemic season, similar results were obtained (Fig D.B in S1 Text).

## Clinical signs and symptoms by serotype

Compared to DENV1, patients with DENV2 had greater odds of ICU admission (Odds Ratio [OR] 2.06, 95% confidence interval [CI]: 1.14–3.98), hemoconcentration (OR = 4.74, 2.04–13.86), low platelet count (OR = 1.61,1.25–2.09 for <100,000 and OR = 2.93, 1.94–4.60 for <50,000), pleural effusion (OR = 2.88, 2.14–3.92), and mucosal bleeding (OR = 1.53,

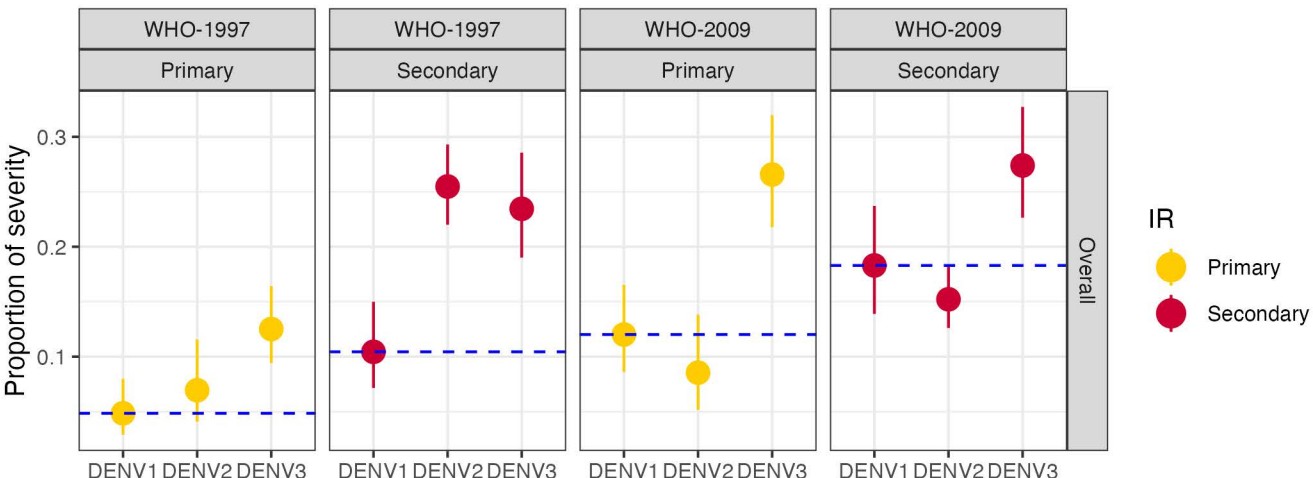

**Fig 3. Model-derived severity by DENV serotype of infection and immune status in the PDCS and PDHS combined, Managua, 2004–2024.** Model-derived estimates compared to DENV1 stratified by WHO disease severity classification and immune status, as indicated. Model was adjusted by age and sex. Due to sample size, DENV4 was excluded from the analysis. Blue dotted line, DENV1 reference value. IR, immune response. P, primary; S, secondary.

1.05–2.28), the last only in secondary infections (Figs 4 and I in S1 Text). We also observed a higher risk of rash (OR = 1.86, 1.49–2.33) and fewer cases reporting headache (OR = 0.45, 0.36–0.57) and myalgia (OR = 0.70, 0.55–0.88) compared to DENV1 (Fig 5). For DENV3, we observed a greater association with general symptoms such as rash (OR = 3.00, 2.36–3.82) and myalgia (OR = 1.60, 1.26–2.03) and alarm signs such as abdominal pain (OR = 2.84, 2.20–3.68) and mucosal bleeding (OR = 1.87, 1.39–2.52) (Fig 5). In terms of severity, DENV3 was correlated with pleural effusion (OR = 2.48, 1.82–3.42), poor capillary refill (OR = 2.05, 1.53–2.77), compensated shock (OR = 2.33, 1.71–3.22), and hypotensive shock (OR = 2.18, 1.45–3.34) compared to DENV1 (Fig 4). See Figs E–J in S1 Text stratified by immune status and study.

Severe dengue cases, defined by the WHO-2009 criteria, were primarily characterized by shock. The WHO-2009 definition encompasses both compensated and hypotensive shock, unlike the WHO-1997 DSS definition, which includes only cases with hypotensive shock. Table 4 depicts patients identified upon admission with either compensated or hypotensive shock and those progressing from compensated to hypotensive shock, regardless of medical intervention. Notably, the severity of DENV1 is characterized by compensated shock, with only 13 (15%) of patients with shock progressing from compensated to hypotensive shock, in contrast to DENV2 and DENV3, where significantly more cases presented with hypotensive shock (31 [24%] and 25 [13%], respectively) or progressed from compensated to hypotensive shock (41 [32%] or 72 [37%], respectively). Further, other symptoms indicative of plasma leakage, such as pleural effusion and ascites, were significantly higher in DENV2 cases (31% and 40%, respectively) and DENV3 cases (25% and 24%, respectively) compared to DENV1 cases (11% and 13%, respectively) (Figs 4, 5 and E–J in S1 Text stratified by immune status and study). ICU admission was significantly higher in DENV2 (11%) and DENV3 (8%) cases and lower in DENV1 cases (5%). Laboratory abnormalities such as hemoconcentration and thrombocytopenia were significantly more common in DENV2 (5% and 42%, respectively) and lower in DENV1 (1% and 25%) and DENV3 (1% and 29%).

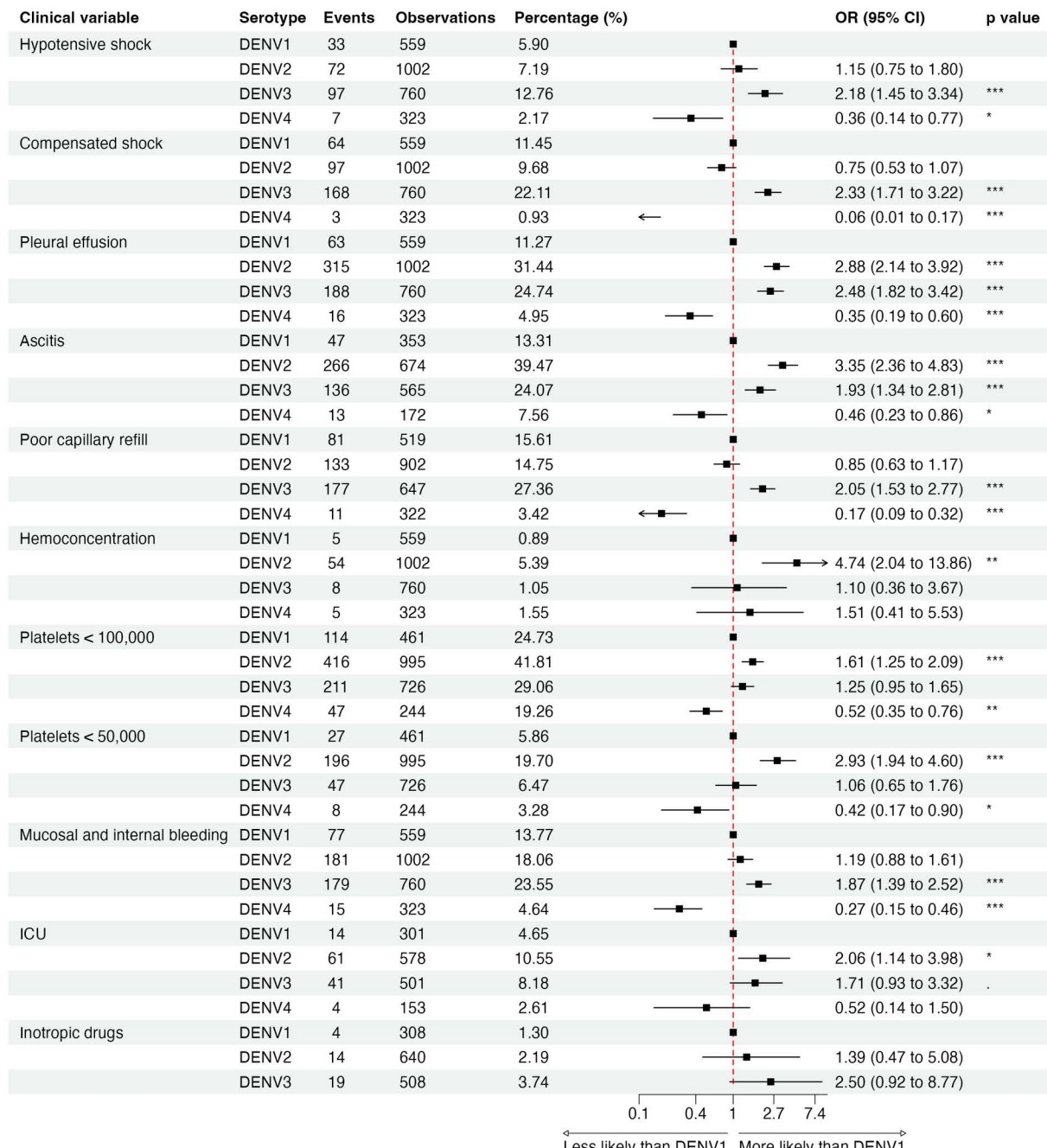

**Fig 4. Clinical signs, clinical laboratory results, and case management of dengue severity by serotype in both primary and secondary cases in the PDCS and PDHS combined, Managua, 2004–2024.** Mucosal bleeding: epitaxis, gingivorrhagia, conjunctival bleeding, hematuria, hematemesis, melena, vaginal bleeding. ICU, intensive care unit.

## Discussion

Dengue virus continues to expand globally, with multiple co-circulating serotypes that cause a wide range of clinical outcomes, and can be potentially fatal. To evaluate how disease severity is modulated by immune status and DENV serotype, we leveraged two complementary long-term studies in Nicaragua, including an integrated follow-up system from primary health

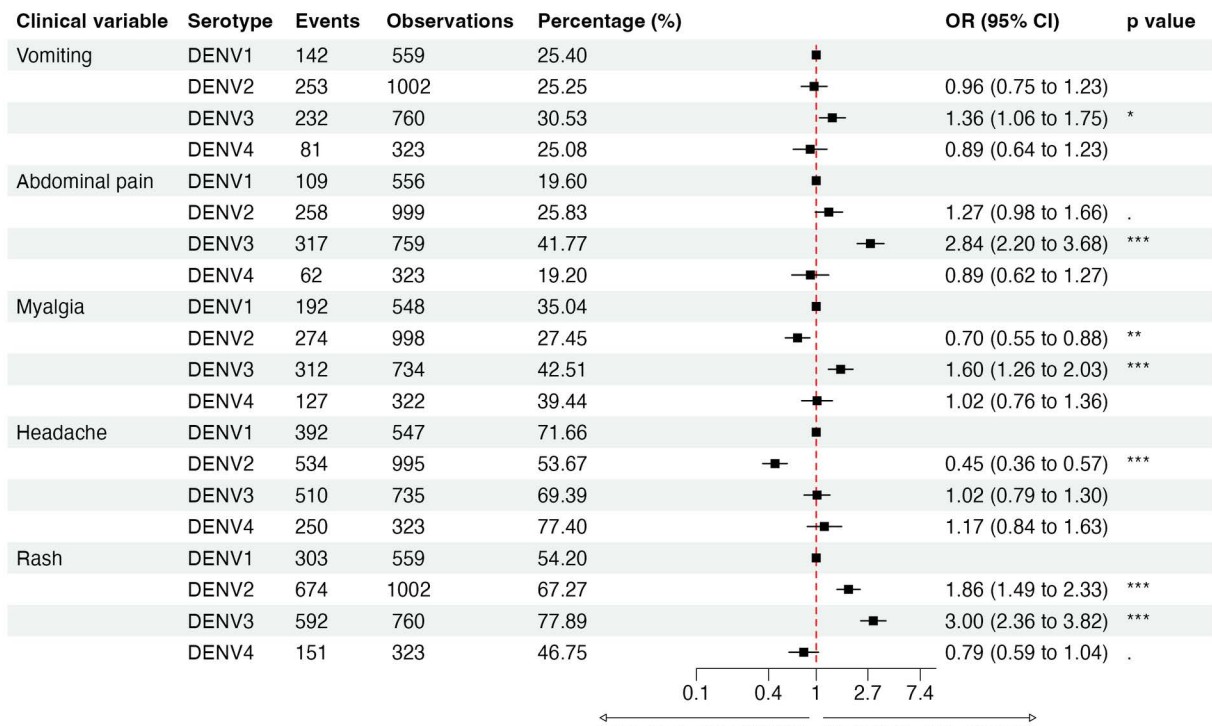

| Clinical variable | Serotype | Events | Observations | Percentage (%) | OR (95% CI) | p value |
|---|---|---|---|---|---|---|
| Vomiting | DENV1 | 142 | 559 | 25.40 | | |
| | DENV2 | 253 | 1002 | 25.25 | 0.96 (0.75 to 1.23) | |
| | DENV3 | 232 | 760 | 30.53 | 1.36 (1.06 to 1.75) | * |
| | DENV4 | 81 | 323 | 25.08 | 0.89 (0.64 to 1.23) | |
| Abdominal pain | DENV1 | 109 | 556 | 19.60 | | |
| | DENV2 | 258 | 999 | 25.83 | 1.27 (0.98 to 1.66) | . |
| | DENV3 | 317 | 759 | 41.77 | 2.84 (2.20 to 3.68) | *** |
| | DENV4 | 62 | 323 | 19.20 | 0.89 (0.62 to 1.27) | |
| Myalgia | DENV1 | 192 | 548 | 35.04 | | |
| | DENV2 | 274 | 998 | 27.45 | 0.70 (0.55 to 0.88) | ** |
| | DENV3 | 312 | 734 | 42.51 | 1.60 (1.26 to 2.03) | *** |
| | DENV4 | 127 | 322 | 39.44 | 1.02 (0.76 to 1.36) | |
| Headache | DENV1 | 392 | 547 | 71.66 | | |
| | DENV2 | 534 | 995 | 53.67 | 0.45 (0.36 to 0.57) | *** |
| | DENV3 | 510 | 735 | 69.39 | 1.02 (0.79 to 1.30) | |
| | DENV4 | 250 | 323 | 77.40 | 1.17 (0.84 to 1.63) | |
| Rash | DENV1 | 303 | 559 | 54.20 | | |
| | DENV2 | 674 | 1002 | 67.27 | 1.86 (1.49 to 2.33) | *** |
| | DENV3 | 592 | 760 | 77.89 | 3.00 (2.36 to 3.82) | *** |
| | DENV4 | 151 | 323 | 46.75 | 0.79 (0.59 to 1.04) | . |

0.1   0.4   1   2.7   7.4

Less likely than DENV1   More likely than DENV1

**Fig 5. Symptoms of dengue by serotype in both primary and secondary cases in the PDCS and PDHS combined, Managua, 2004–2024.**

**Table 4. Compensated shock and hypotensive shock by DENV serotype in patients with shock.**

| Serotype | Only compensated shock | Only hypotensive shock | Hypotensive from compensated shock | p-value* |
|---|---|---|---|---|
| DENV1 | 51 (61%) | 20 (24%) | 13 (15%) | – |
| DENV2 | 56 (44%) | 31 (24%) | 41 (32%) | 0.03 |
| DENV3 | 96 (50%) | 25 (13%) | 72 (37%) | <0.001 |
| DENV4 | 2 (22%) | 6 (67%) | 1 (11%) | NA |

*Chi-square test, NA = Not Applicable.

center to hospital, based on 19 years of data. We used both the 1997 and 2009 WHO severity classification schemes, revealing important distinctions. Overall, we found that DENV2 and DENV3 caused the highest percent of DHF/DSS, and DENV3 caused the highest percent of SD, as well as the greatest number of severe clinical manifestations. DENV2 severity occurred primarily in secondary cases, while DENV3 induced severity in primary and secondary infections. This is consistent with previous publications [9,10,15–17,32–35] and is concerning given that currently approved vaccines have poor efficacy against DENV2 for Dengvaxia and DENV3 for QDenga vaccines in dengue-naïve recipients [7,8].

When analyzing immune status by serotype, we found that secondary cases predominated for DENV2 and DENV4, whereas more similar numbers of primary and secondary cases were observed for DENV1 and DENV3, again supported by the literature [18,32,36,37]. Even in younger age groups, DENV2 and DENV4 showed a higher percentage of secondary cases, suggesting that primary infections by these serotypes were asymptomatic, consistent

with previous reports [9,16,32,38,39]. Likewise, severe disease occurred much more often in secondary DENV2 and DENV4 cases, in contrast to DENV1 and DENV3 severe cases, which were more evenly distributed among primary and secondary cases, as reported elsewhere as well [9,17,36,38–42].

We analyzed the clinical spectrum of dengue considering the two distinct WHO severity classifications of 1997 and 2009. The 1997 definition is more focused on the pathophysiology of vascular leak leading to shock, while the 2009 definition aims for real-time identification of clinical disorders to enable timely case management [20]. The clinical spectrum of dengue using the WHO-1997 definition revealed that most dengue cases evolved as DF, regardless of the serotype involved, with approximately 16% being classified as DHF/DSS. In contrast, using the 2009 classification, only 24% of cases evolved as DwoWS, with the majority as DwWS, and 17% classified as SD. Sixty-two percent of the patients classified as DHF/DSS (WHO-1997) were classified as DwWS (WHO-2009). All exhibited a clinical profile characterized by plasma leakage and thrombocytopenia that did not progress to shock or respiratory distress.

Breaking this down by serotype and immune status, secondary cases showed a higher proportion of WHO-1997 severe disease (DHF/DSS), especially in the case of DENV2 for DSS, while DENV3 had a high proportion of DHF/DSS in secondary cases as well as primary cases. Using the WHO-2009 classification, the serotype that exhibited the highest proportion of SD cases was DENV3 for both primary and secondary infections. Overall, DENV2 was associated with severity using the 1997 classification, whereas DENV3 exhibited more severe cases according to both 1997 and 2009 definitions. It is important to note that the WHO-1997 DSS definition includes only cases with hypotensive shock, unlike the WHO-2009 SD definition, which encompasses both compensated and hypotensive shock.

With respect to clinical signs and symptoms, severe thrombocytopenia (<50,000 platelets) was five times more prevalent in DENV2 than in DENV3 cases. Other markers of severity such as hemoconcentration, a sign of plasma leakage, were more common among DENV2 cases, underscoring the importance of this serotype. Both DENV2 and DENV3 were associated with other signs of plasma leakage, while DENV3 was significantly associated with internal and mucosal bleeding.

Of the WHO-2009 SD cases, two-thirds were not classified as severe by WHO-1997, because they failed to meet the full criteria for DHF/DSS. These were predominantly DENV3 cases (49%) followed by DENV2 (25%), DENV1 (22%) and DENV 4 (4%), almost entirely characterized by some form of shock without thrombocytopenia or hemorrhagic manifestations. Interestingly, most severe cases of DENV1 did not progress to hypotensive shock, maintaining only hemodynamic alterations such as poor capillary refill without advancing to hypotension and/or decreased pulse pressure. This is why DENV1 appears to be more severe than DENV2 when applying the 2009 classification. The severity of both DENV2 and DENV3 is characterized by signs of plasma leakage, resulting in ascites and pleural effusion. However, in comparison with DENV1, DENV3 is significantly associated with hemodynamic alterations leading to shock, either in its initial, compensated stage or during the hypotensive phase. Surprisingly, DENV2 did not have this association with shock, whether compensated or hypotensive, when compared to DENV1. This discrepancy may be attributed to a marked difference in the severity of DENV2 when analyzing immune status, where primary cases evolve favorably.

Previous reports have also identified that DENV2 (WHO-1997) and DENV3 (WHO-1997 and WHO-2009) account for the greatest dengue severity [9,15–17]. A meta-analysis of 170 global dengue outbreaks from 1990 to 2015 ranked DENV2 as the most frequent cause of outbreaks, followed by DENV1, with DENV3 and DENV4 less commonly reported [43]. Importantly, the mortality rates associated with DENV2 infections were the highest, followed

by DENV3, DENV4, and DENV1 rates. These findings suggest that despite their lower frequency, DENV3 outbreaks are associated with significant disease and mortality.

Our findings have implications for dengue vaccines. Severe dengue can occur in infections by each of the four serotypes; in our study, the most severity was observed in DENV2 and DENV3 infections. Further, severe disease can occur in secondary infections by all serotypes and in primary DENV1 and DENV3 infections. However, currently registered vaccines provide imbalanced protection; Dengvaxia did not provide protection against any serotype in seronegative participants and showed most efficacy against DENV4 [7,44]. Qdenga is a strong DENV2 vaccine but does not protect against DENV3 in seronegative recipients, and there was not enough information in the clinical trials on protection against DENV4 [8]. A vaccine that provides balanced protection continues to be an important public health need. Beyond serotype, it is important to consider genotype, as vaccine efficacy can differ by genotype and lineage [45,46]. The findings from this study are specific to the circulating genotypes in Nicaragua (DENV1 GV; DENV2 GIII; DENV3 GIII; DENV4 GII), and clinical presentation may be different with other genotypes. Thus, future research is needed to analyze severity by serotype with other genotypes and lineages.

One of the key strengths of our study lies in the comprehensive assessment of dengue severity by serotype, utilizing the two WHO classifications from 1997 and 2009. This aspect is particularly important given that most of the literature on dengue severity utilizes the 1997 classification. The introduction of different serotypes into Nicaragua and their varying impacts on disease severity based on which classification is used highlights the importance of future research presenting results using both classification approaches. Another strength is the longitudinal nature of the study with a large number (almost 3,000) of laboratory-confirmed pediatric dengue cases and a single protocol in place for over almost two decades with consistency of clinical care and data collection. Finally, we analyzed community-based and hospital-based cases, as well as those who progressed from the health center to the hospital – made possible by over a decade of careful work by our physicians harmonizing clinical variables between the two sites and a dedicated team of programmers and data managers creating our customized informatics system and algorithm pipelines.

One limitation of our study was the short circulation time of DENV4, which prevented us from comprehensively analyzing the clinical signs and symptoms of severity in this serotype. Additionally, at the time of this analysis, one genotype of each serotype had circulated in Managua during the study period. Thus, we were not able to compare severity across genotypes. We plan to expand this analysis following the introduction of new lineages of serotypes 1, 3, and 4 in 2022 [45]. Another potential limitation is that our studies are primarily focused on children, since historically, children have been the most affected population in Nicaragua, as in many regions worldwide. This may reduce the generalizability of our findings to the adult population. However, our studies have documented an overall DENV seroprevalence of 70% at 10 years of age, progressively increasing to 83% by age 17. Specifically, in this study, 1,725 dengue patients were secondary cases, representing 65.6%. We hypothesize that these children, who have previously encountered DENV, could exhibit immunological responses similar to those of adults. Nevertheless, we still consider that differences may exist, as dengue epidemiology in Nicaragua suggests that adults may experience more than two DENV infections, which could influence the progression of disease. Additionally, adults have more comorbidities that can impact dengue disease severity (e.g., obesity, diabetes, hypertension) [47,48].

Overall, our study reveals that DENV2 and DENV3 consistently lead to the highest severity in dengue cases with certain differences according to WHO classification schema. Further, DENV2 primarily impacts secondary cases, while DENV3 induces severity in primary

and secondary infections. These findings highlight the complex interplay between DENV serotype, immune status and the classifications of disease severity. Moreover, they underscore the need for a vaccine that offers balanced efficacy against all four serotypes.

## Supporting information

**S1 Text.  Table A.** Classification of dengue disease severity by 1997 and 2009 World Health Organization criteria and definition of clinical variables. **Fig A. DENV circulation in the PDCS and PDHS combined, Managua, 2004–2024.** Circulating serotypes in dengue cases in the studies in Managua, Nicaragua. **Fig B. DENV circulation by immune status stratified by study.** Upper panel, Pediatric Dengue Cohort Study; lower panel, Pediatric Hospital-based Study in Managua, Nicaragua, 2004–2024. Circulating serotypes and immune status in Managua are represented. The hospital-based study was not conducted in 2020 and 2021 due to COVID-19. **Fig C. Distribution by immune response among severe dengue cases, stratified by study, Managua, 2004–2024.** Upper panel, Pediatric Dengue Cohort Study (Cohort); lower panel, Pediatric Dengue Hospital-based Study (Hospital). IR, immune response. **Fig D. Severity by infecting DENV serotype and immune status by study, Managua, 2004–2024.** A) Upper panel, Pediatric Dengue Cohort Study (Cohort); lower panel, Pediatric Dengue Hospital-based Study (Hospital). B) Severity by infection of DENV serotype adjusted for epidemic seaon. Due to sample size, DENV-4 was excluded from the analysis. IR, immune response. **Fig E. Clinical signs, clinical laboratory results, and case management of dengue severity by serotype in primary DENV infections in the Pediatric Dengue Hospital-based study, 2004–2024.** A) Clinical signs, laboratory results, and clinical management. B) Symptoms of dengue. **Fig F. Clinical signs, clinical laboratory results, and case management of dengue severity by serotype in primary DENV infections in the PDCS and PDHS combined, Managua, 2004–2024.** A) Clinical signs, laboratory results, and clinical management. B) Symptoms of dengue. **Fig G. Clinical signs, clinical laboratory results, and case management of dengue severity by serotype in primary DENV infections in the Pediatric Dengue Cohort Study, 2004–2024.** A) Clinical signs, laboratory results, and clinical management. B) Symptoms of dengue. **Fig H. Clinical signs, clinical laboratory results, and case management of dengue severity by serotype in primary DENV infections in of the Pediatric Dengue Hospital-based study, 2004–2024.** A) Clinical signs, laboratory results, and clinical management. B) Symptoms of dengue. **Fig I. Clinical signs, clinical laboratory results, and case management of dengue severity by serotype in secondary DENV infections in the PDCS and PDHS combined, Managua, 2004–2024.** A) Clinical signs, laboratory results, and clinical management. B) Symptoms of dengue. **Fig J. Clinical signs, clinical laboratory results, and case management of dengue severity by serotype in primary DENV infections in the Pediatric Dengue Cohort Study, 2004–2024.** A) Clinical signs, laboratory results, and clinical management. B) Symptoms of dengue.
(DOCX)

## Acknowledgments

We are grateful to Cesar Narvaez for his contribution to data management for the Pediatric Dengue Hospital-based Study. We thank both past and current team members, based at the Hospital Infantil Manuel de Jesús Rivera, Sócrates Flores Vivas Health Center, the Laboratorio Nacional de Virología in the Centro Nacional de Diagnóstico y Referencia, and the Sustainable Sciences Institute in Nicaragua for their commitment and exceptional work. We are deeply grateful to the participants of the Pediatric Dengue Cohort Study and Pediatric Dengue Hospital-based Study and their families.

The findings and conclusions in this report are those of the author(s) and do not necessarily represent the views of the Centers for Disease Control and Prevention.

## Author contributions

**Conceptualization:** Federico Narvaez, Daniel Camprubí-Ferrer, Gabriela Paz Bailey, Eva Harris, Angel Balmaseda.

**Data curation:** Carlos Montenegro, José Victor Zambrana, Sonia Arguello, Angel Balmaseda.

**Formal analysis:** Jose G. Juarez, José Victor Zambrana, Daniel Camprubí-Ferrer.

**Funding acquisition:** Eva Harris, Angel Balmaseda.

**Investigation:** Federico Narvaez, Karla Gonzalez, Fanny Barrios, Sergio Ojeda, Miguel Plazaola, Nery Sanchez, Guillermina Kuan, Angel Balmaseda.

**Methodology:** Carlos Montenegro, Jose G. Juarez, José Victor Zambrana.

**Project administration:** Elsa Videa, Eva Harris, Angel Balmaseda.

**Resources:** Eva Harris.

**Supervision:** Federico Narvaez, Elsa Videa, Fanny Barrios, Sergio Ojeda, Miguel Plazaola, Nery Sanchez, Guillermina Kuan, Eva Harris, Angel Balmaseda.

**Validation:** Federico Narvaez, Carlos Montenegro, Jose G. Juarez, José Victor Zambrana, Sonia Arguello, Daniel Camprubí-Ferrer, Gabriela Paz Bailey.

**Visualization:** Jose G. Juarez, José Victor Zambrana.

**Writing – original draft:** Federico Narvaez, Jose G. Juarez, José Victor Zambrana, Daniel Camprubí-Ferrer, Gabriela Paz Bailey, Eva Harris, Angel Balmaseda.

**Writing – review & editing:** Federico Narvaez, Carlos Montenegro, Jose G. Juarez, José Victor Zambrana, Karla Gonzalez, Elsa Videa, Sonia Arguello, Fanny Barrios, Sergio Ojeda, Miguel Plazaola, Nery Sanchez, Guillermina Kuan, Gabriela Paz Bailey, Eva Harris, Angel Balmaseda.

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
