## [Decision Letter · Decision Letter 0]

9 Oct 2024

Dear Dr. Balmaseda,

Thank you very much for submitting your manuscript "Dengue severity by serotype and immune status in 19 years of pediatric clinical studies in Nicaragua" for consideration at PLOS Neglected Tropical Diseases. As with all papers reviewed by the journal, your manuscript was reviewed by members of the editorial board and by several independent reviewers. The reviewers appreciated the attention to an important topic. Based on the reviews, we are likely to accept this manuscript for publication, providing that you modify the manuscript according to the review recommendations. 

There are suggestions from the reviewers that can improve the importance of the manuscript and some corrections or modifications.

1) As the incidence severe dengue shifts to older ages, it is important to comment how this data can be extrapolated to older patients

2) To investigate correlation of the incidence with the severe cases with the level of a circulating DENV serotype in a season. If appropriate, apply some adjustment to consider for this potential bias.

3) one other point is to expand the discussion about the implications of these findings in different epidemiological settings. In general vaccinologists and government officers have not consider the differences in pathogenicity among serotypes, and sequence of infection in analyzing their vaccine safety and efficacy

Sincerely,

Ernesto T. A. Marques, M.D./Ph.D

Academic Editor

Elvina Viennet

Section Editor

There are suggestions from the reviewers that can improve the importance of the manuscript and some corrections or modifications.

1) As the incidence severe dengue shifts to older ages, it is important to comment how this data can be extrapolated to older patients

2) To investigate correlation of the incidence with the severe cases with the level of a circulating DENV serotype in a season. If appropriate, apply some adjustment to consider for this potential bias.

3) one other point is to expand the discussion about the implications of these findings in different epidemiological settings. In general vaccinologists and government officers have not consider the differences in pathogenicity among serotypes, and sequence of infection in analyzing their vaccine safety and efficacy

Reviewer's Responses to Questions

**Key Review Criteria Required for Acceptance?**

**Methods**

-Are the objectives of the study clearly articulated with a clear testable hypothesis stated?

-Is the study design appropriate to address the stated objectives?

-Is the population clearly described and appropriate for the hypothesis being tested?

-Is the sample size sufficient to ensure adequate power to address the hypothesis being tested?

-Were correct statistical analysis used to support conclusions?

-Are there concerns about ethical or regulatory requirements being met?

Reviewer #1: The manuscript has well defined objectives, totally aligned with the methodology to test the hypothesis.

The size of the samples is actually very good and allows to achieve more of the results. Nevertheless, the cohort studied only considered child up to 17 yo; these might represent some bias, and it should be explained if these results may be extended to adults where the impact of dengue transmission might be higher. 

Statistical methods are adequate according to the objectives.

Any concern regarding ethical or regulatory issues was identified.

Reviewer #2: The manuscript entitled “Dengue severity by serotype and immune status in 19 years of pediatric clinical studies in Nicaragua” by Narvaez et al. investigates the relationship of viral serotype and pre-existing immunity to dengue with dengue disease severity in children aged 6 months-17 years old.

The manuscript explores a very relevant topic in dengue disease and has a detailed description of the data and the analysis used for reaching the conclusions. Although this topic has been previously explored in several studies in the dengue field, the main strengths of this study are the comparison of both WHO classifications of dengue severity, from 1997 and 2009, and the analysis of 2,630 community-based and hospital-based cases over a period of almost two decades.

**Results**

-Does the analysis presented match the analysis plan?

-Are the results clearly and completely presented?

-Are the figures (Tables, Images) of sufficient quality for clarity?

Reviewer #1: All the results are clearly expressed and analyzed. Tables and figures are adequate.

Reviewer #2: Some points to address are listed below.

-Please, clarify the age of participants. In line 32, participant age is 6 months-17 years old; in lines 50 and 97, participant age is 2-17 years old; and in table 1, age group <1 years is included.

-In lines 173 to 175, the prevalence analysis of primary and secondary infections according to age group (Table 2) does not include the statistical significance. I suggest including the p-values.

-In lines 182-186, the prevalence analysis of primary and secondary infections according to disease severity caused by each serotype also does not include the statistical significance (Table 3 and Figure 2). Likewise, I suggest including the p-values.

- Why are two studies included in Supplementary Figs. 2 and 3 (Pediatric Dengue Cohort Study and Pediatric Dengue Hospital-based Study) instead of three studies, as indicated in Table 1 and Supplementary Figs 4-9? Please clarify.

-Did the comparative analysis of community-based and hospital-based cases yield any difference in clinical signs and symptoms of dengue and laboratory results?

-Figures 4 and 5 and Supplementary Figs 4-9. The number of events and observations from Supplementary Figures 4-9 (stratified by immune status and study) do not match those from Figures 4 and 5. Please clarify.

For example, in Fig 4 there are 33 events of hypotensive shock in 557 DENV-1 infections, while in Supplementary Figs 4-9 there are 66 events of hypotensive shock in 1114 DENV-1 infections (considering primary and secondary infections and the three studies). The same happens with the other clinical variables.

-In lines 207-213, the analysis of compensated and hypotensive shock by DENV serotype according to data from Table 4 should include the statistical significance of the results, indicating the p-values.

**Conclusions**

-Are the conclusions supported by the data presented?

-Are the limitations of analysis clearly described?

-Do the authors discuss how these data can be helpful to advance our understanding of the topic under study?

-Is public health relevance addressed?

Reviewer #1: Although the conclusions are coherent and well discussed, I think there are some limitations to be addressed. The cohort is limited to child ranging less than 1 to 17 yo; for different reasons, dengue incidence is usually higher in young-adult populations, who actually might had been infected more than once (or several times), therefore the results may (or may not) be different. Also, it is not clear how the circulation of the different DENV serotypes were by the time of the sampling; if we have a predominant circulation of DENV3, it is expected that more of the severe cases are for DENV3. This information might be important to enrich the discussion.

The public health relevance is established. One of the most important issues is in regards of the vaccines that differentially protect for different serotypes. Perhaps, having a comment on how to tackle this issue would be interesting.

Reviewer #2: (No Response)

**Editorial and Data Presentation Modifications?**

Reviewer #1: Only minor revision is required and will be explained in the general comments.

Reviewer #2: Minor comments:

-Table 1, include “Age (years)” and “Sex” (instead of sexo) in the first column.

-Fig. 1, the maximum value on the y-axis scale should be included (600).

-Table 3, correct “DWW” to “DWWS” in the first column.

**Summary and General Comments**

Reviewer #1: The manuscript and the work is solid; the size of the cohort is impresive, and allows a lot of good quality data to be collected and analyzed.

There are some limitations that I think should be considered and discussed. The cohort is limited to child ranging less than 1 to 17 yo; for different reasons, dengue incidence is usually higher in young-adult populations, who actually might had been infected more than once (or several times), therefore the results may (or may not) be different. Some discussion to explain how these results may or may not be extended to other populations, including possible different proportions on severity might be useful.

On the other hand, it is not clear how the circulation of the different DENV serotypes were by the time of the sampling; if we have a predominant circulation of DENV3, it is expected that more of the severe cases are for DENV3. Perhaps a map or a table indicating the proportion of each serotype circulating through the time might be interest and will facilitate the data analysis.

Finally, one of the most relevant conclusions are about the vaccines and the unbalanced protection against different serotypes. Nevertheless, the discussion around this issue is vague, and more can be elaborated in terms of how we can expect and impact if the vaccination is massive in different settings (high, medium and low endemicity, for instance) and some ideas on how this issue can be (eventually) addressed.

Reviewer #2: (No Response)

PLOS authors have the option to publish the peer review history of their article (what does this mean? ). If published, this will include your full peer review and any attached files.

**Do you want your identity to be public for this peer review?** For information about this choice, including consent withdrawal, please see our Privacy Policy .

Reviewer #1: No

Reviewer #2: No

Figure Files:

While revising your submission, please upload your figure files to the Preflight Analysis and Conversion Engine (PACE) digital diagnostic tool, https://pacev2.apexcovantage.com . PACE helps ensure that figures meet PLOS requirements. To use PACE, you must first register as a user. Then, login and navigate to the UPLOAD tab, where you will find detailed instructions on how to use the tool. If you encounter any issues or have any questions when using PACE, please email us at figures@plos.org.

Data Requirements:

Please note that, as a condition of publication, PLOS' data policy requires that you make available all data used to draw the conclusions outlined in your manuscript. Data must be deposited in an appropriate repository, included within the body of the manuscript, or uploaded as supporting information. This includes all numerical values that were used to generate graphs, histograms etc.. For an example see here: http://www.plosbiology.org/article/info%3Adoi%2F10.1371%2Fjournal.pbio.1001908#s5 .

Reproducibility:

References

---

## [Editor Report · Decision Letter 1]

25 Dec 2024

Dear Dr. Balmaseda,

We are pleased to inform you that your manuscript 'Dengue severity by serotype and immune status in 19 years of pediatric clinical studies in Nicaragua' has been provisionally accepted for publication in PLOS Neglected Tropical Diseases.

Best regards,

Elvina Viennet, PhD

Section Editor

Elvina Viennet

Section Editor

Shaden Kamhawi

co-Editor-in-Chief

Paul Brindley

co-Editor-in-Chief
